# Glycoprotein 5-Derived Peptides Induce a Protective T-Cell Response in Swine against the Porcine Reproductive and Respiratory Syndrome Virus

**DOI:** 10.3390/v16010014

**Published:** 2023-12-21

**Authors:** Fernando Calderon-Rico, Alejandro Bravo-Patiño, Irasema Mendieta, Francisco Perez-Duran, Alicia Gabriela Zamora-Aviles, Luis Enrique Franco-Correa, Roberto Ortega-Flores, Ilane Hernandez-Morales, Rosa Elvira Nuñez-Anita

**Affiliations:** 1Facultad de Medicina Veterinaria y Zootecnia, Universidad Michoacana de San Nicolas de Hidalgo, Km. 9.5 S/N carretera Morelia-Zinapecuaro, La Palma, Tarimbaro PC 58893, Mexico; 1028153g@umich.mx (F.C.-R.); alejandro.bravo@umich.mx (A.B.-P.); 0618713g@umich.mx (F.P.-D.); 1106849c@umich.mx (A.G.Z.-A.); 1837980k@umich.mx (L.E.F.-C.); 0840020c@umich.mx (R.O.-F.); 2Posgrado en Ciencias Quimico-Biológicas, Facultad de Quimica, Universidad Autonoma de Queretaro, Cerro de las Campanas S/N, Querétaro PC 76010, Mexico; alicia.mendieta@uaq.mx; 3Escuela Nacional de Estudios Superiores Unidad Leon, Universidad Nacional Autonoma de Mexico, Blv. UNAM No. 2011, Leon PC 37684, Guanajuato, Mexico; ihernandez@enes.unam.mx

**Keywords:** PRRS, pig virus, T-cells, CD8+, GP5, epitopes

## Abstract

We analyzed the T-cell responses induced by lineal epitopes of glycoprotein 5 (GP5) from PRRSV to explore the role of this protein in the immunological protection mediated by T-cells. The GP5 peptides were conjugated with a carrier protein for primary immunization and booster doses. Twenty-one-day-old pigs were allocated into four groups (seven pigs per group): control (PBS), vehicle (carrier), PTC1, and PTC2. Cytokine levels were measured at 2 days post-immunization (DPI) from serum samples. Cytotoxic T-lymphocytes (CTLs, CD8+) from peripheral blood were quantified via flow cytometry at 42 DPI. The cytotoxicity was evaluated by co-culturing primed lymphocytes with PRRSV derived from an infectious clone. The PTC2 peptide increased the serum concentrations of pro-inflammatory cytokines (i.e., TNF-α, IL-1β, IL-8) and cytokines that activate the adaptive cellular immunity associated with T-lymphocytes (i.e., IL-4, IL-6, IL-10, and IL-12). The concentration of CTLs (CD8+) was significantly higher in groups immunized with the peptides, which suggests a proliferative response in this cell population. Primed CTLs from immunized pigs showed cytolytic activity in PRRSV-infected cells in vitro. PTC1 and PTC2 peptides induced a protective T-cell-mediated response in pigs immunized against PRRSV, due to the presence of T epitopes in their sequences.

## 1. Introduction

Porcine reproductive and respiratory syndrome virus (PRRSV) is one of the most devastating viruses in the pork industry worldwide [1,2,3]. PRRSV belongs to the family *Arteriviridae* in the order *Nidovirales.* It is an enveloped single-stranded positive-sense RNA virus [4,5]. The PRRSV genome is polycistronic [6], of approximately 14.9 kb to 15.5 kb in length. It contains at least 10 open reading frames (ORFs) [7,8], namely, ORF1a, ORF1b, ORF2a, ORF2b, ORF3, ORF4, ORF5, ORF5a, ORF6, and ORF7 [9].

Currently, there are two main PRRSV genotypes: type 1, or the European genotype (PRRSV EU-type), also known as the Lelystad virus (LV); and type 2, known as the American genotype (NA-type), also called VR-2332 [4,10]. In 2006, a new genotype emerged: the highly pathogenic PRRSV (HP-PRRSV), originating from PRRSV type 2 [11]. During the HP-PRRSV outbreak in China, HP-PRRSV caused over 400,000 deaths and affected 2 million pigs [3,11]. HP-PRRSV causes severe lung injury, characterized by extensive hemorrhage, inflammatory cells, and serous fluid infiltration in the lungs’ vascular system [12].

Particularly, PRRSV type 2 causes severe reproductive impairment in sows and respiratory pathologies in pigs of all ages [11]. PRRSV is frequently related to the porcine respiratory disease complex (PRDC), which is promoted by poor sanitary practices, stressful environments, and opportunistic viruses and bacteria [13,14].

In 2013, worldwide PRRSV-related economic losses were estimated at USD 664 million [15,16]. Current estimations indicate losses of USD 2.5 billion in the USA and Europe alone [17,18]. Therefore, the swine industry is highly affected by PRRSV worldwide.

Vaccination strategies can control the spread of PRRSV because they reduce the viral load in blood [19] and reduce tissue damage in target organs, such as the alveoli. Hence, viral load reduction is an important correlate of vaccine protection [19]. Vaccination, however, does not induce sterilizing immunity. Therefore, viral transmission still occurs [18].

Commercial PRRS-modified live virus (MLV) vaccines are the most widely used strategy for disease control, but they provide partial or no protection against heterologous viruses [19,20,21]. Other commercial vaccines include inactivated virus vaccines or strain-specific vaccines (i.e., autologous vaccines manufactured with virus isolated from the same farm) [22,23]. The current vaccines fail to provide effective protection because PRRSV possesses several evasion strategies and high antigenic heterogeneity [24].

Vaccine efficacy is improved by adjuvants, which is the reason for developing novel adjuvants [25]. The most common adjuvants are aluminum salts (commercially known as Alhydrogel^®^), which are widely used because they have been approved by the FDA based on their safety and efficacy [26].

New-generation vaccines use subunits, synthetic or recombinant proteins, and DNA or RNA [27]. For PRRSV, the vaccines in development include subunit vaccines and virus-vectored vaccines [28]. Nevertheless, more research is needed to improve the effectiveness of PRRSV vaccine candidates against heterologous strains [29]. For instance, studies have confirmed that GP5 peptides induce specific antibodies but do not correlate with antibody-mediated neutralization [30,31]. Therefore, GP5 is an interesting target for vaccine development.

The first studies using vaccines based on baculovirus-expressed GP3, GP5, and N subunits did not show the induction of IFN-producing cells. The GP3 and GP5 subunits showed 50 to 68% protection against PRRSV, whereas the N protein did not. In general, these types of vaccines show similar results compared to the commercial inactivated vaccines. Based on this, it could be considered that protein subunit vaccines are not effective against PRRSV [29]. Antibody titers are the classical correlates of protection for PRRSV vaccine candidates. In this context, GP5-derived peptides have been studied as neutralization epitopes, particularly linear peptides [32]. Peptides can bind to MHC and activate the immune system [33]; they are also capable of activating dendritic cells (DCs), promoting DC maturation and T-cell activation in vitro, stimulating DC migration, and the priming of functional T-cells in vivo [34].

The evaluation of the protective response provided by cellular immunity is not commonly used as a correlate of protection. Therefore, the efficacy of the cellular immunity mediated by cytotoxic lymphocytes (CTLs) is poorly understood. This cellular immunity has been recently explored for vaccine candidates against SARS-CoV-2 and influenza viruses [35,36,37].

For PRRSV, epitopes identified from a contemporary PRRSV strain exhibited high immunostimulatory activity that improved the cellular responses mediated by CD4+ helper T-lymphocytes and CD8+ cytotoxic T-lymphocytes [38]. Peptides with the potential to stimulate cytotoxic cells are conserved among PRRSV type 2 strains [39]. In this regard, cytotoxic T-lymphocytes are key effector cells because they can induce the apoptosis of PRRSV-infected cells [39]. This information supports the hypothesis that GP5-derived peptides, containing putative T epitopes, may induce a cytotoxic T-cells response against PRRSV [40].

Consequently, the objective of this study was to demonstrate the contribution of CTL to the protective immunity induced by a peptide derived from GP5 of PRRSV, as has been demonstrated for other viruses such as SARS-CoV-2 and influenza [35,36,37].

## 2. Materials and Methods

### 2.1. Peptides

The peptides were named PTC1 and PTC2 and are derived from the GP5 sequence of PRRSV type 2. The PTC1 peptide was previously identified in our work group. The antigenicity of PTC1 was determined via bioinformatic analysis (Immunoepitope Data base IEDB and CCL Main Workbench 20.0). In contrast, the PTC2 peptide contains two epitopes together reported by Vashit et al., 2008 [40]. Moreover, PTC1 was chosen due to its high degree of conservation (over 80% homology) among the different genotypes reported in the database.

Both peptide sequences match the endodomain of GP5, corresponding to residues 117 to 163. The peptides were chemically synthesized by solid-phase synthesis by GenScript (Piscataway, NJ, USA). PTC1, RLYRWRSPVIGHLIDLKRVVRVSAEQWGRP (30 aa) and PTC2, KGRLYRWRSPVIVEKLAALICFVIRLAKN (29 aa), showed 95 and 98% purity, respectively. To bind peptides with a carrier protein, we included a cysteine at the terminal carboxyl end. Both peptides were resuspended in dimethyl sulfoxide (DMSO) and adjusted to a concentration of 1 mg/mL of a stock solution. The peptides were used at a concentration of 4 g/mL for the detection of specific antibodies and immunoreactivity assays.

We compared the sequences of PTC1 and PTC2 with the sequences reported for PRRSV type 1, 2, and high pathogenic strains in the Protein BLAST program. The results show a Query Cover for PTC1 of 100% for PRRSV type 1, 96–100% for PRRSV type 2, and 93–100% for HP-PRRSV. PTC2 showed a Query Cover of 100% for the three genotypes. These results suggest that the sequences are conserved [41]

### 2.2. Animal studies

#### Experimental Design, Animals, and Housing

The pigs originated from a certified farm (Good Livestock Practices or BPP) where the animals were regularly tested and found to be negative for PCV2 and PRRSV. We selected randomly twenty-one-day-old, weaned piglets from six litters without distinguishing their sex. The piglets were a crossbreed of Large White and Pietrain. They were all subjected to a standard preventive medicine plan, which included iron supplementation (iron zinc) and vaccinations. The vaccinations included *Coccidia* (Coxizuril 5%, 20 mg/Kg orally, Lapisa, Mexico) administered 3 days after birth, *Mycoplasma hyopneumoniae* and PCV 2 (Comboflex 2.0 mL IM) administered at 3 days and 19 days after birth, and swine influenza (Porcimune influenza 2.0 mL IM Lapisa, Mexico) administered at 21 days old.

At the age of 21 days, we transferred 28 piglets to a separate experimental stockyard measuring >33.6 m^2^. Each piglet was ear-tagged and randomly assigned to one of two groups, with 14 individuals in each group. To determine the assignment, we used the GraphPad-Random number generator online tool, which randomly distributed the animals into four groups of seven individuals each. 

On day zero of the experiment, the animals were inoculated based on their assigned group. The control group received 800 µL of PBS (N = 7), the vehicle group received a preparation of 400 µL of PBS and 400 µL of aluminum hydroxide (adjuvant) (N = 7), the PTC1 group received a preparation containing PTC1 (N = 7), and the PTC2 group received a preparation containing PTC2 (N = 7). The inoculation preparation for the peptides consisted of 200 µg of peptide PTC1 or PTC2 (1 mg/mL), 200 µg maleimide [1 mg/mL] (Thermofisher Maleide Activated BSA), conjugated according to manufacturer specifications, and 400 µL adjuvant alhydrogel (InvivoGen, San Diego, CA, USA) with a final volume of 800 µL. The piglets were immunized intramuscularly by attaching their legs. Hereafter, the control and vehicle groups will be referred to as the control groups.

We visually examined the pigs daily and weighed them weekly. The bedding was cleaned daily, and the piglets had ad libitum access to drinking water and were fed a commercial diet (Folechon phase 1, Vitalechon phase 1, phase 2, phase 3, and Vitalechon initiator Valleys) following the dosages recommended by the manufacturer, using a semi-automatic device (baby pig feeder of 9 pounds). In addition, the stockyards were enriched with play material (suspended chains and bottles). The animals were given a seven-day adaptation period before the start of the experiment. The immunization and blood sampling schedule is shown in (Appendix A).

Animal housing, handling, animal care, and sampling were conducted under the guidelines of the Animal Care and Use Program (NIH, Washington, DC, USA) and the Official Mexican standard NOM-062-ZOO-1999. The study protocol CICUMSNH-A101-FMVZ was approved by the Institutional Research Ethics Committee (Committee on Animal Research and Ethics of the Facultad de Medicina Veterinaria y Zootecnia, from Universidad Michoacana de San Nicolas de Hidalgo).

### 2.3. Blood Sampling and Isolation of Peripheral Blood Mononuclear Cells

Blood samples were collected via jugular vein puncture at 2 and 42 days after treatment. The blood was collected with and without EDTA (15 mL per animal) using Vacutainer tubes (BD, Franklin Lakes, NJ, USA) and processed within 3 h of collection.

Peripheral blood mononuclear cells (PBMC) were isolated from whole blood using density gradient separation with Lymphoprep™ (StemCell Technologies, Vancouver, BC, Canada) according to the manufacturer’s standard procedure. Briefly, 15 mL complete blood samples were diluted 1:1 (volume:volume) with Lymphoprep reagent, then centrifugated at 464× *g* for 25 min. The interface was collected and transferred to a conical tube, washed two times with 2 mL of PBS, and then centrifugated at 694× *g* for 15 min. RBC lysis buffer 1X (Milltenyi) was added and incubated at room temperature for 5 min. Then, washed two times with 2 mL of PBS, and centrifuged at 573× *g* during 5 min. The cell pellet was resuspended in 2400 µL of FBS/DMSO 5% and divided into four cryotubes. The samples were deep-frozen at −80 °C until being used for subsequent analyses.

### 2.4. Cytokine Detection Assay Panel

The cytokine concentration was quantified using the commercial pre-configured immunoassay multiplex Cytokine and Chemokine 9-Plex Porcine ProcartaPlex™ panel kit (Invitrogen, Vienna, Austria, Cat. EPX090-60829-901), following the manufacturer’s protocols (ProcartaPlex™ Multiplex Immunoassay user guide). To that end, we used the Luminex 200 instrument (Austin, TX, USA). The concentration was evaluated in serum samples obtained two days after immunization (2 DPI). The cytokines assessed included IFN-α, IFN-γ, IL-1β, TNF-α, IL-4, IL-6, IL-8, IL-10, and IL-12p40. When the Luminex instrument indicated a concentration “below the limit of detection,” the values were reported as zero.

### 2.5. Cytometric Analysis

Cytometry was conducted using approximately 1 × 10^6^ cells/mL of PBS per sample. The cells were stained and analyzed using the Attune NxT acoustic focusing cytometer (Thermo Fisher Scientific, Carlsbad, CA, USA), which is equipped with a blue laser (488 nm) and a violet laser (405 nm). The analysis focused on three subpopulations of T-cells: CD3+/CD8+/CD44+, CD8+/CD44−, and CD8−/CD44+. The antibodies used for staining were anti-pig-CD3ε PE-Cy™7 (Becton Dickinson, cat: 561477), anti-pig-CD8α FITC (Invitrogen, cat: MA5-28714), and anti-pig-CD44 PE (Invitrogen, cat: MA1-19781).

A minimum of 10,000 events in the cells gate were collected for each sample to analyze the individual event packages. A dot plot was generated by comparing forward scatter-A against forward scatter-H, and any doublets were excluded. A linear dot plot of side scatter against linear forward scatter was used to identify the lymphocyte population. Within the lymphocyte population, a subset gating strategy was employed to compare BL1-CD3+ cells against the linear scatter side. Then, another subset quadrant gating was performed to select four subpopulations of T-lymphocytes: CD8−/CD44− (Q4), CD8+/CD44+ (Q2), CD8+/CD44− (Q3), and CD8−/CD44+ (Q1) (Appendix A). The results obtained from each subpopulation are presented as a percentage of positive cells.

### 2.6. In Vitro Assays

#### Plasmid and Cell Lines

The infectious clone of North American PRRSV type 2 (strain NVSL 97-78950) cloned in the plasmid pFL-12 was generously provided by Dr. Asit K. Pattnaik of the University of Nebraska. The cell line PK-15 Siglec-10+ CD163+ (PK-15 S10-CD163) was kindly donated by Dr. Hans J. Nauwynck of the University of Ghent. PK-15 S10-CD163 and MARC-145 (ATCC CRL-2378.1) cells were cultured in Dulbecco’s Modified Eagle Medium (DMEM) supplemented with 10% fetal bovine serum (FBS), 1% penicillin/streptomycin (P/S; 10,000 U/mL; 10 mg/mL; Thermo Fisher Scientific, Carlsbad, CA, USA), and 0.5% gentamycin (10,000 U/mL; Thermo Fisher Scientific).

### 2.7. In Vitro Transcription and Virus Recovery

The plasmid pFL-12 containing the full genome of PRRSV type 2 was quantified using the NanoDrop™ 2000 (Thermo Fisher Scientific, Carlsbad, CA, USA). Endpoint PCR was performed with the primers previously described by our research group: ORF3 5-3′: forward: GGATGTACCGCACCATGGAA, reverse: CCTGTCATGCGCAGATTGTG (pb: 187); ORF4 5-3′: forward: AAGGCCACTTGACCAGTGTT, reverse ATGCTGCGGTAGTGTTGGTT: (pb: 368); ORF5 5-3′: forward: TGCTCGCGATTGCTTTCTTT, reverse: ATGAGAGCTGCTGTTGCTGT (pb: 84); ORF6 5-3′: forward: CCTTGACACAGTCGGTCTGG, reverse: AGGACATGCAGTTCTTCGCA (pb: 140) and ORF7 5-3′: forward: GTTGGGTGGCAGAAAAGCTG, reverse: GGGGCTTCTCCGGGTTTTA (pb: 210). Gel electrophoresis was subsequently performed using the complete plasmid and the amplifications of the ORFs to confirm the presence and integrity of the plasmid and PRRSV genome.

Next, the plasmid was linearized with the AclI enzyme (New England Biolabs, Ipswich, MA, USA) for RNA in vitro transcription, which was carried out using the TranscriptAid T7 High Yield kit (Thermo Fisher Scientific, Carlsbad, CA, USA) according to the manufacturer’s instructions. Then, 5 µg of viral RNA was transfected into MARC-145 cells using Lipofectamine 3000™ (Invitrogen, Thermo Fisher Scientific, Carlsbad, CA, USA) as previously described [42]. The infected MARC-145 cells were incubated for two days, after which the cells and supernatant were lysed. A passage was then made into a new MARC-145 culture with DMEM 10% FBS without antibiotics and incubated during for 48 h to allow virus replication. The cells and the supernatant were subsequently lysed, aliquoted, and stored at − 80 °C in the original DMEM media supplemented with 20% FBS.

### 2.8. RNA Extraction, Retro Transcription, Endpoint PCR, and Real-Time PCR Assay for Virus Detection

Viral RNA was extracted from 500 μL of MARC-145 cells infected with PRRSV, as described previously. RNA extraction was carried out using 500 μL of TRIzol™ reagent (Invitrogen Thermo Scientific, Carlsbad, CA, USA), following the user guide for RNA isolation. The RNA pellet was then resuspended in 15 μL of RNase-free water and kept on ice. Reverse transcription was performed using the RevertAid H Minus Reverse Transcriptase™ kit (Thermo Fisher Scientific, Carlsbad, CA, USA) following the manufacturer’s instructions.

The viral copies were quantified via qPCR with 11 µL of cDNA and the Syber green™ reagent (Biorad, Hercules, CA, USA), following the manufacturer’s recommendations. The primers used were ORF7 forward 5′–3′: GTTGGGTGGCAGAAAAGCTG and reverse 5′–3′: GGGGCTTCTCCGGGTTTTA, 0.5 µL [10 µM], which amplify a product of 210 pb from ORF7.

### 2.9. Standard Curve for PRRRSV Quantification

The qPCR standard curve was performed using the pFL-12 PRRSV plasmid. The concentration of the plasmid was correlated with an estimated viral copy number [43]. Six-fold serial dilutions ranging from 6.4 × 10^9^ to 2.0 × 10^6^ in viral copy number were used to measure viral copies in serum samples and infected cells (Appendix A).

The calculation of the standard curve parameters showed a correlation coefficient (R2) of 0.996, a slope of −3.159, an intercept of 21.55, and an efficiency (E) of 107.279%. The viral copy number in the examined samples could be quantified by applying the standard formula for the regression analysis (Y = −3.159X + 21.55). 

### 2.10. Co-Cultures of Primary Lymphocytes with PK-15 S10-CD163 Cells Infected with PRRSV

The cytotoxicity assay was performed using a co-culture of PK-15 S10-CD163 as target T-cells and primary porcine lymphocytes isolated from the experimental groups as primed cells.

Initially, the PK-15 S10-CD163 cells were seeded at a concentration of 10,000 cells/well in a 96-well black plate with a clear flat bottom (Thermo Scientific, Carlsbad, CA, USA) in DMEM 2% FBS (Thermo Scientific, Carlsbad, CA, USA). The cells were incubated for 24 h at 37 °C and 5% CO_2_. Subsequently, the cells were infected with PRRSV at a multiplicity of infection (MOI) of 0.2 in DMEM 10% inactivated FBS and seeded simultaneously with the lymphocytes in the plate.

The primary lymphocytes cryopreserved from 42 DPI were thawed at room temperature and seeded into 100 mm culture plates (Corning, NY, USA) under standard culture conditions using RPMI (Thermo Scientific, Carlsbad, CA, USA) 10% FBS. Prior to the experiment, the viable lymphocytes were normalized as a control for viability. Viability was measured using a 3-(4,5-dimethylthiazol-2-yl)-2,5-diphenyltetrazolium bromide (MTT) assay. The cells were cultured at a density of 1 × 10^4^ cells/well in RPMI basal medium and incubated at 37 °C in 96-well plates with 5% CO_2_. Then, 20 µL of MTT (5 mg/mL in PBS) was added to each well, and cells were incubated for 4 h at 37 °C. After centrifugation at 830× *g* for 5 min, the supernatants were immediately removed. Then, 100 µL of dimethyl sulfoxide (DMSO) was added, and the absorbance (595 nm) was measured using a microplate reader (Bio-Rad, Hercules, CA, USA). Cell viability was expressed as percentage of viability compared to fresh lymphocytes as the control. The viability percentage was calculated as follows: [(optical density of the samples)/(optical density of the control)] × 100. The samples were analyzed in triplicate. All groups showed approximately 40% viability after thawing (Appendix A).

After incubation, the non-attached lymphocytes were collected and identified as the lymphocyte isolate. These lymphocytes were divided into two samples: one was cultured with RPMI 10% FBS, and the other was cultured with RPMI 10% FBS supplemented with 10 µg/mL of phytohemagglutinin (PHA) (SIGMA Aldrich, St. Louis, MO, USA). The cells were incubated for 24 h at 37 °C and 5% CO_2_.

For the co-culture assays, lymphocyte isolate was added to the infected PK-15 cells. In detail, primary lymphocytes were added at a concentration of 8000 cells/50 µL/well to achieve an 8:1 ratio (lymphocyte: PK-15), using RPMI 10% FBS without phenol red (Thermo Scientific, Carlsbad, CA, USA). Additionally, each well received 50 µL of Green celltox reagent (Promega, Madison, WI, USA). The co-cultures were incubated at 37 °C with 5% CO_2_. The fluorescence was quantified at 12 h and 24 h using a Varioskan LUX (Thermo Scientific, Carlsbad, CA, USA) instrument measured at 485/520 nm.

### 2.11. Statistical Analysis

The statistical differences between the infected and non-infected groups were initially assessed using normality and lognormality tests for all assays. Subsequently, one-way or two-way analysis of variance (ANOVA) was conducted, followed by post hoc tests such as Kruskal–Wallis or Brown–Forsythe and Welch post hoc tests, as specified in the figure legends. Significance levels of * *p* < 0.05, ** *p* < 0.01, and *** *p* < 0.001 were considered statistically significant. The data are presented as means ± standard error of the mean (SEM). The statistical software GraphPad Prism (version 8.0.1, GraphPad Software, San Diego, CA, USA) was utilized for the analysis.

## 3. Results

### 3.1. Weight Gain Was Not Influenced by the Immunization Protocol

Weekly weight gain increased steadily over the course of the experiment in the four groups. There were non-significant differences observed between the PBS and vehicle groups compared to the PTC1 and PTC2 groups (Figure 1).

### 3.2. PTC2 Peptide Increased the Concentration of Proinflammatory Cytokines

Cytokine concentrations were quantified in serum samples collected 2 DPI post-immunization to determine the proinflammatory state. Pigs immunized with PTC2 exhibited significantly higher serum concentration of TNF-α and IL-1β compared to both the control and vehicle groups (Figure 2A,B). In contrast, there were no significant differences in the serum concentrations of IL-8 and IFN-α between the P TC2 group and the control groups (Figure 2C,D). The PTC1 group did not show differences in cytokine serum concentrations compared to the controls. These findings suggest that the PTC2 peptide induces a proinflammatory condition after immunization. The cytokine concentrations were measured 48 h after immunization to observe the immunomodulatory effect during the initial stage of antigen processing and presentation. 

### 3.3. PTC2 Peptide Increased the Serum Concentration of Cytokines Related to the Activation of Cellular Immunity

Pigs in the PTC2 group exhibited a significant increase in their serum concentrations of IL-4, IL-10, and IL-6 compared to both the control and vehicle groups (Figure 3A,B,D,E). Additionally, the concentration of IL-12 increased in the vehicle, PTC1, and PTC2 groups compared to the control group (Figure 3C). These findings suggest that the vehicle used in this experimental setup may influence the concentration of IL-12.

### 3.4. PTC1 and PTC2 Immunizations Correlated with Increased CD8+ Cell Count in Peripheral Blood

The quantification of cytotoxic lymphocytes (CD8+) in peripheral blood was conducted 42 days after immunization. The results show that the pigs immunized with PTC1 and PTC2 exhibited a higher CD8+ cell population compared to the groups treated with PBS and the vehicle (Figure 4A). Although the group immunized only with the vehicle depicted a higher CD8+ cell count, it was still lower than the count observed in the pigs immunized with both peptides. These findings suggest that the peptides may have a direct effect on the production and differentiation of cytotoxic lymphocytes. 

To determine whether the peptides could influence the activation state of cytotoxic lymphocytes, we quantified CD8+/CD44+ double-positive cells via flow cytometry. However, no significant difference in CD8+/CD44+ count was observed between the peptide-immunized groups compared to the control groups (Figure 4B).

### 3.5. PTC1 Primed Lymphocytes Recognize and Eliminate PRRSV Infected Cells

Lymphocytes isolated at 42 DPI from all experimental groups were co-cultivated with PRRSV-infected cells to evaluate their cytotoxic activity. The cytotoxic activity was estimated based on the relative fluorescence units (RFUs), which were indicative of non-viable cells. As a viability control, PK-15 cells (alone) infected with the virus were used.

After 12 h of co-incubation, there was not a significant difference in viable cells among the treatment groups, indicating a similar level of cell death regardless of the immunization scheme (Figure 5A). In contrast, when lymphocytes were stimulated with PHA, the group immunized with PTC1 showed a significantly higher difference in cell viability (Figure 5B).

After 24 h of coincubation, the effect of PTC1 on cell viability was notably higher than that of PTC2 and the control groups, both with and without PHA stimulation (Figure 5C,D).

## 4. Discussion

Subunit-based PRRSV vaccine candidates have been studied for over 25 years. The protective efficacy of these vaccines varies and is comparable to that of inactivated virus vaccines [29]. MLV vaccines are currently the most widely used type, but these vaccines do not provide complete and sterilizing protection against PRRSV, nor do other new-generation vaccines that are currently under investigation. Consequently, there is an ongoing need for effective and safe PRRSV vaccines. Notably, both natural and synthetic peptides have demonstrated a robust induction of T-cell mediated immunity [44], thereby indicating their potential promise for vaccine development.

In this study, we evaluated peptides from GP5 as potential vaccines candidates. First, the experimental setup did not affect the health or growth rate of the pigs. In our study, weight gain was independent of the immunization protocol (Figure 1), suggesting that the pigs maintained a normal growth rate. Frequently, monitoring weight gain serves as an indicator of the health and welfare status of production animals [45]. Furthermore, all the piglets used for the experiment were confirmed to be PRRSV-negative, as determined using PCR (Appendix A).

We report here the immunogenic characteristics of two PRRSV type 2 GP5 peptides as potential vaccine candidates, with a particular focus on their ability to induce T-cell mediated immunity. PTC1 encompasses residues 148–161 (Seq. Ref. UJD73100.1), and PTC2 corresponds to residues 116–130 and 148–163 (Seq. Ref. UOA04012.1). PTC1 shares 76% similarity with the sequence previously reported by Chen, C. et al. [46]. The GP5-149T contains a T-cell epitope and has been shown to induce a T-cell immune response [46]. PTC2 contains two sequences; the first one corresponds to an immunodominant T-cell epitope reported by Vashisht, K et al. [40]. The second sequence overlaps the PTC1 sequence. T-cell epitopes have been identified in GP5, which supports the use of this protein to induce a T-cell mediated response [40,46,47,48]. Studies have found that peptides similar to PTC1 and PTC2 have induced the activation of IFN-γ secreting cells (CTLs) and proliferation of PBMCs in mice and pigs in assays related to T-cells [40,46,48].

The vaccine candidate formulation, containing one of two peptides derived from GP5, showed different stimulation patterns concerning proinflammatory cytokines, cytokines related to the activation of cellular immunity, and the cytotoxic activity of primed lymphocytes.

The proinflammatory profile of the pigs immunized with the peptides was evaluated by measuring the concentration of TNF-α, IL1-β, IL-8, and IFN-α (Figure 2). The results demonstrate an increase in TNF- α and IL1-β (Figure 2A,B), suggesting that PTC1 induces a proinflammatory state at 2 DPI. Similar studies using peptides derived from PRRSV and carrier proteins as a vehicle revealed an increase in TNF- α, a cytokine associated with the augmentation of the Th1 immune response [46]. The addition of carrier proteins to the peptides has proven to be an effective strategy to enhance the immune response to vaccine antigens [46].

To assess the profile of cytokines related to the activation of cellular immunity, we focused our cytokines measurement at 48 h, as previous reports have indicated changes during the first 24 and 48 h post-treatment. The changes involved an increase in pro-inflammatory cytokines and signaling pathways related to T-cell activation [47,49]. Following this, it is observed that signaling pathways such as STAT, which is associated with the Th1 phenotype, decreased [47]. Arunachalam et al., 2023 reported that after 3 days of vaccination with the Pfizer-BioNTech BNT162b2 vaccine, cytokines like CXCL10, IL-6, IFN-α, and IFN-γ returned to normal levels [50,51].

Immunization with PTC2, after 2 DPI or 48 h, corresponded to an elevated serum concentration of cytokines related to the Th1 and Th2 immunological response (Figure 3). Specifically, the higher concentration of IFN-γ and IL-12 is associated with the activation of helper lymphocytes of the Th1 lineage [52], whereas the higher concentration of IL-4 and IL-6 suggests the stimulation of Th2 lymphocytes [53,54].

This study reveals that cellular immunity associated with Th1 and Th2 was activated. This same response has been observed in a murine model immunized with SARS-CoV-2 [55]. Th1/Th2-related immune responses through the secretion of their predominant cytokines such as IFN-ɣ and IL-4 could confer protection [55]. The activation of Th1/Th2 may contribute to the reduction of viral loads and could also significantly prevent body weight loss, as observed in SARS-CoV-2. In addition, the potential secretion of neutralizing antibodies is favored with the activation of Th1 and Th2 [55]. In our study, the role of neutralizing antibodies was not examined, but in future research, the secretion of neutralizing antibodies will be investigated to determine their ability to recognize and neutralize PRRSV.

Overall, PTC2 induces a T helper Th1/Th2 proinflammatory cytokine immunomodulatory profile, whereas PTC1 primes lymphocytes (likely CTLs) with a cytotoxic effect on PRRSV-infected cells.

The cytotoxic activity of the CTL response has been previously studied using peptides derived from type 1 and HP-PRRSV [56,57]. In these studies, pigs immunized with NSP2 and M-protein-derived peptides demonstrated an increased activity of CTLs producing IFN-γ [56], suggesting that this response could be sufficient to protect against PRRSV, even in the absence of neutralizing antibodies [57]. Moreover, it has been suggested that peptides capable of inducing CTLs might be required for optimal immunity against PRRSV [58,59,60].

In our study, we evaluated the cytotoxic lymphocyte population in pigs immunized with PTC1 and PTC2 peptides at 42 DPI. Both peptides induced a significant increase in CD8+. Previously, in a mice model, we observed an increase in the CTL population (CD8+) after PTC1 immunization. This result is confirmed in the present study, where we observed an increased population of total cytotoxic lymphocytes (CD8+) with both the PTC1 and PTC2 groups compared to the control and vehicle group (Figure 4). The increase in total cytotoxic lymphocytes in the vehicle group (BSA protein) is well characterized for the adjuvant aluminum hydroxide (alhydrogel), because it modulates the T-lymphocyte response by increasing the inflammasome receptor protein 3 (NLRP3, also called NALP3) as well as other inflammasome-independent pathways. These pathways promote the secretion of high levels of proinflammatory factors through antigen-presenting cells. These particles act directly or indirectly on B- and T-cells [45]. Moreover, aluminum salts increase the population of activated cytotoxic cells [61], promote the cross-presentation of DC antigens in vitro, and improve the cytolytic activity of CTLs [61,62].

We evaluated the activation phenotype of CD8+ T-cells (CD8+/CD44+) at 42 DPI. Despite observing a higher population in the immunization groups, there was no significant difference in the activation profiles of the CD8+ population. These results align with the findings of Franzoni et al., 2013, who demonstrated that a viral challenge is needed to observe a difference in CD8+ activation. Specifically, only the group immunized with an attenuated Classical Swine Fever Virus (CSFV) vaccine and subsequently challenged with a virulent strain depicted a higher CD8+ activation profile after 56 days of immunization [63]. This result suggests that primed CD8+ could be activated by the presence of the viral antigen in vitro or in vivo after a correct stimulation in the immunized pigs.

Given that the CD8+ cell population was higher in immunized pigs, we further explored their cytotoxic activity in vitro to confirm that the peptides could prime lymphocytes in our experimental setup. We conducted in vitro cytolysis co-culture assays based on previously reported methodologies [64]. CD8+ from immunized pigs showed increased cytotoxic activity when co-cultured with PRRSV-infected cells and stimulated with PHA (Figure 5). This suggests that PTC1 and PTC2 may prime CD8+ cells upon immunization. It is well established that PHA can induce lymphocyte proliferation through the cross-linking of glycoproteins on the cell surface [65,66]. PHA can activate T-lymphocytes as it binds to the TCR/CD3 complex, mimicking all intracellular activation events, similar to the mechanism triggered by anti-CD3 antibodies [67].

The activity of specific CTLs is a promising indicator of vaccine protection. For instance, Y. Li et al. (2023) showed that primed CTLs enhance protection against PRRSV type 1 in a transplacental infection context, reducing viral load in infected animals [68].

Finally, all these data suggest that both peptides are effective in inducing a cellular immune-mediated response. PTC2 elicited a cytokine reaction characteristic of a coordinated Th1/Th2 response, while PTC1 promoted cytolytic activity. The mechanisms by which PTC1 and PTC2 induce such immune response remain to be elucidated.

## 5. Conclusions

PTC1 and PTC2 peptides induce a T-cell response and a pro-inflammatory profile in immunized pigs against PRRSV. This suggests a potent and effective immune response, likely attributable to the presence of T epitopes within their sequences. 

## Figures and Tables

**Figure 1 viruses-16-00014-f001:**
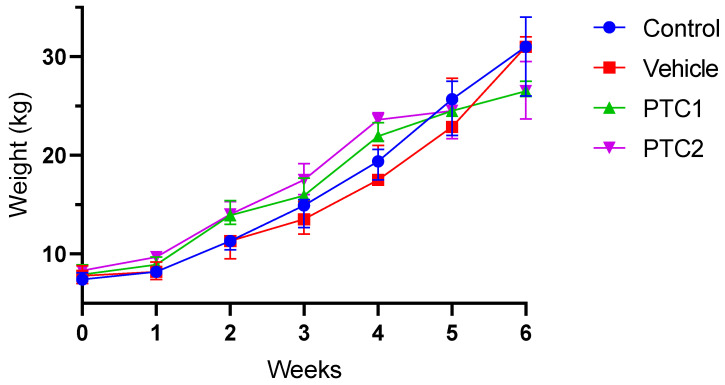
Weight gain. Weight of pigs at different growth stages (N = 28) divided into four groups (N = 7). The curves represent the median weight of seven individual subjects, while the error bars indicate the interquartile range. Statistical significance was determined using a Two-way ANOVA and Tukey pos hoc test. Non-significant differences were observed.

**Figure 2 viruses-16-00014-f002:**
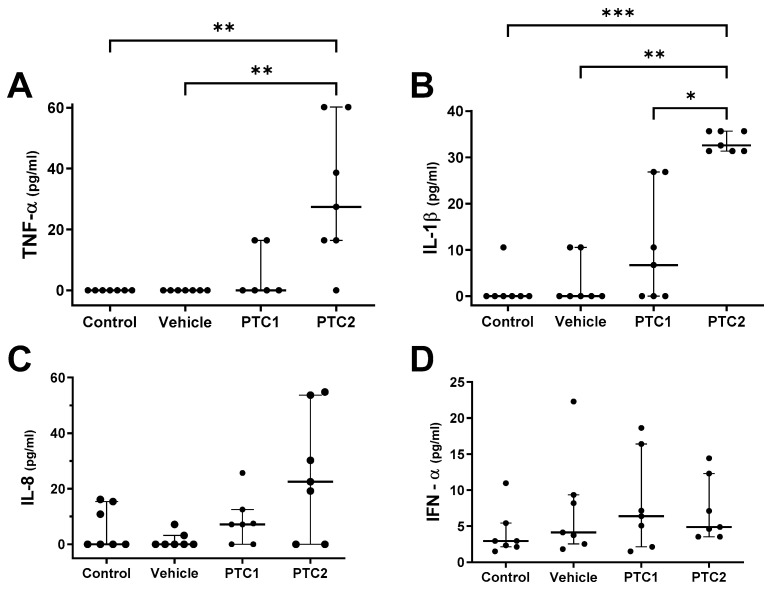
Analysis of proinflammatory cytokines in serum from pigs immunized with PTC1 and PTC2. Serum samples were collected 48 h post-inoculation. Cytokines were measured using a multiplex Luminex-based cytokine immunoassay. (**A**) TNF-α. (**B**) IL-1b. (**C**) IL-8. (**D**) IFN-α. Bars indicate the median of 7 individual subjects and the error bars represent the interquartile range. Statistical significance was determined using ANOVA and Kruskal–Wallis post hoc test. Significance levels of * *p* < 0.05, ** *p* < 0.01, and *** *p* < 0.001 were considered statistically significant. ns: non-significant.

**Figure 3 viruses-16-00014-f003:**
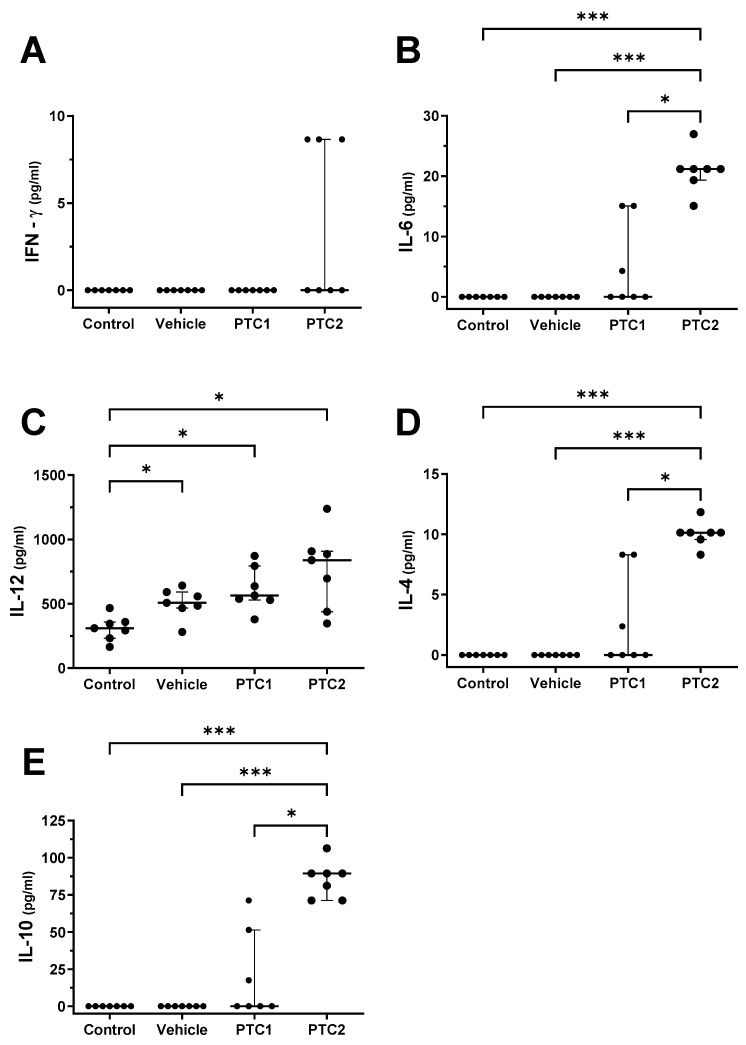
Quantification of IFN-γ, IL-6, IL-12, IL-4, and IL-10 in pig sera. Sera from pigs were collected at 48 h post inoculation. Cytokines were measured using a multiplex Luminex-based cytokine immunoassay. Bars indicate the median of 7 individual subjects and the error bars represent the interquartile range. (**A**) IFN-γ, (**B**) IL-6, (**C**) IL-12, (**D**) IL-4, and (**E**) IL-10. Statistical significance was determined using ANOVA and Kruskal–Wallis post hoc test. Significance levels of * *p* < 0.05 and *** *p* < 0.001 were considered statistically significant. ns: non-significant.

**Figure 4 viruses-16-00014-f004:**
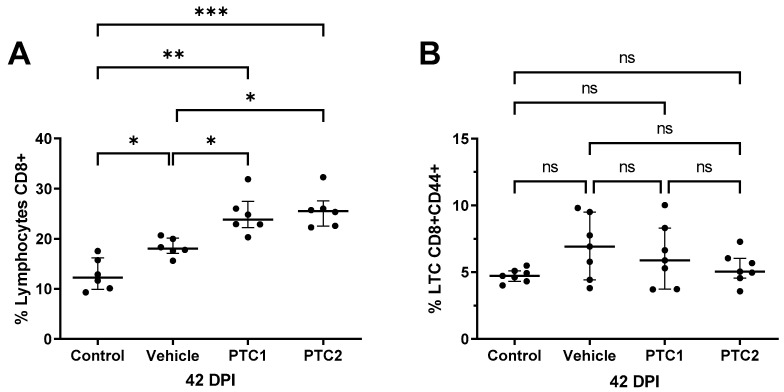
Analysis of T-lymphocyte, CD8+, and CD8+CD44+ cell populations from pigs at 42 DPI. (**A**) Percentage of total cytotoxic cell population CD8+ and (**B**) percentage of total activated cytotoxic cells CD8+CD44+. Bars indicate the median of 7 individual subjects and the error bars represent the interquartile range. Statistical significance was determined using ANOVA and Brown–Forsythe and Welch post hoc tests. Significance levels of * *p* < 0.05, ** *p* < 0.01, and *** *p* < 0.001 were considered statistically significant. ns: non-significant.

**Figure 5 viruses-16-00014-f005:**
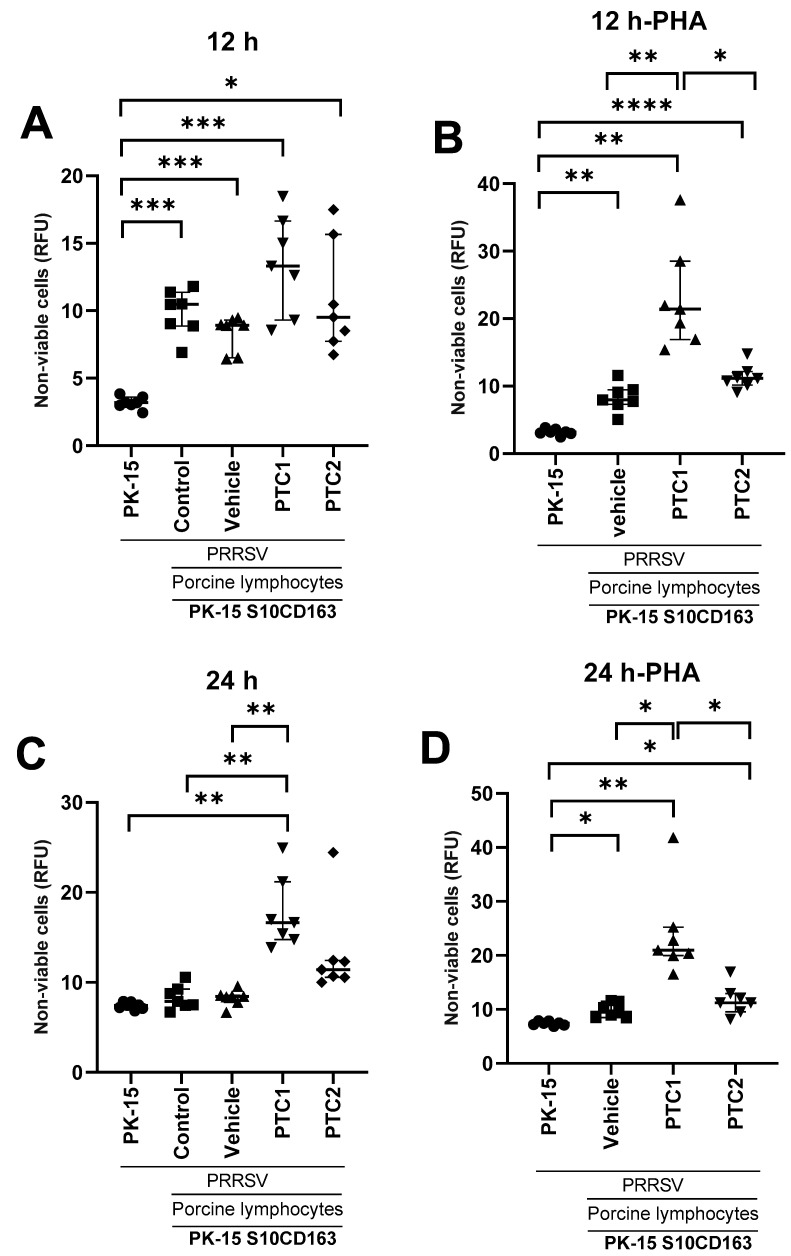
PRRSV peptide-specific CTL responses in vitro. PK-15 S10-CD163 cells were seeded at a ratio of 1:8 with isolated lymphocytes from immunized pigs according to the experimental groups: control, vehicle, PTC1, and PTC2. Additionally, lymphocytes from the vehicle, PTC1, and PTC2 were pre-activated with PHA in vitro for 24 h before co-cultivation. (**A**) RFU measurement at 12 h of co-culture. (**B**) RFU measurement at 12 h of co-culture with pretreated lymphocytes with PHA. (**C**) RFU measurement at 24 h of co-culture. (**D**) RFU measurement at 24 h of co-culture with pretreated lymphocytes with PHA. Bars indicate the median of 7 individual subjects, and the error bars represent the interquartile range. Statistical significance was determined using ANOVA and Brown–Forsythe and Welch post hoc tests. Significance levels of * *p* < 0.05, ** *p* < 0.01, *** *p* < 0.001, and **** *p* < 0.0001 were considered statistically significant. ns: non-significant.

## Data Availability

Data are contained within the article.

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
