# Peer review of "Glycoprotein 5-Derived Peptides Induce a Protective T-Cell Response in Swine against the Porcine Reproductive and Respiratory Syndrome Virus"

_viruses, 2023, doi:10.3390/v16010014_

Round 1

Reviewer 1 Report

Comments and Suggestions for Authors

It is significant to analysize the T cell response induced by lineal epitopes of GP5 from PRRSV and this will be the basic data for developing the new type vaccines. it is an interesting story on peptides induced T cell response about PRRSV.

1. In Fig. 4, when the CD8+ lymphocytes showed the higher percentage in PTC1 and PTC2 groups, why the CD8+CD44+ lymphocytes showed the lower percentage in the same groups?

2. In Fig 5, PRRSV peptide-specific CTL responses was showed at two timepoints, It can not show the role of PTC2 because it has not obvious difference in all the groups.

3.The logic of writing in introduction and discussion can be improved.

Author Response

Reviewer 1

It is significant to analyze the T cell response induced by lineal epitopes of GP5 from PRRSV and this will be the basic data for developing the new type vaccines. It is an interesting story on peptides induced T cell response about PRRSV.

Q1. In Fig. 4, when the CD8+ lymphocytes showed the higher percentage in PTC1 and PTC2 groups, why the CD8+CD44+ lymphocytes showed the lower percentage in the same groups?

Response

This is because the population of CD8+ cells correspond to the general population of cytotoxic lymphocytes circulating in peripheral blood and we wanted to observe if it was possible for this population, in general to increase after immunization. However, we went further and quantified the subpopulation of activated cytotoxic lymphocytes that correspond to CD8+ and CD44+ double positive cells. In this last result, we did not observe significant changes, as mentioned in the discussion (line 516-517), it seems that an antigen such as the complete virus could be required to trigger the increase of this population.

As we mentioned in the text: the results of Fig. 4 B (line 516-517) "are in agreement with Franzoni et al. 2013, where it is evidenced that a viral challenge is needed to observe a difference in CD8+ activation. "

Q2. In Fig 5, PRRSV peptide-specific CTL responses was showed at two timepoints, It cannot show the role of PTC2 because it has not obvious difference in all the groups.

Response

Precisely, we agree with the reviewer. In the work we emphasize that the cytotoxic activity under these conditions is favored only by the role of PTC1 peptide, line 408 “the effect of PTC1 on cell viability was clearly higher than PTC2”, and discussion line 542 “PTC1 promoted cytolytic activity”. But the role of PTC2 cannot be elucidated from this experiment, because it induces proinflammatory cytokines and activators of Th helper lymphocytes. This indicates that the response to PTC2 is mainly towards a cytokine reaction, lines 541-542 “PTC2 triggered a cytokine reaction characteristic to that of a coordinated Th1/Th2 response”.

Q3.The logic of writing in introduction and discussion can be improved.

Response

The introduction and discussion sections were reviewed and improved.

Reviewer 2 Report

Comments and Suggestions for Authors

In this study, Fernando Calderón-Rico and olleagues evaluated the protective T cell response induced by GP5-derived peptides in pigs. This study is well-designed and the manuscript is acceptable for publication after the below problems are solved.

1. Please analyze the conservation of PTC1 and PTC2 in different strains of PRRSV.

2. Why not select multiple time points to analyze the changes of cytokines after immunization?

3. Why are challenge studies not being conducted to determine the effectiveness of vaccines in preventing the disease?

Author Response

Reviewer 2

In this study, Fernando Calderón-Rico and colleagues evaluated the protective T cell response induced by GP5-derived peptides in pigs. This study is well-designed, and the manuscript is acceptable for publication after the below problems are solved.

Q1. Please analyze the conservation of PTC1 and PTC2 in different strains of PRRSV.

Response

We compared the sequences of PTC1 and PTC2 with the sequences reported for PRRSV type 1, 2, and high pathogenic strains in the Protein BLAST program. The results showed a Query Cover for PTC1 of 100% for PRRSV type 1, 96-100% for PRRSV type 2 and 93-100% for PRRSV type HP. PTC2 showed a Query Cover of 100% for the three genotypes. These results suggested that the sequences are conserved.

These results were added to the methodology section where the peptides were described.

Q2. Why not select multiple time points to analyze the changes of cytokines after immunization?

Response

That would be indeed an interesting result, however, for this first approach with the pig model, (the final individual where this biotechnological development will be applied) we wanted to investigate the early response of cytokines. The innate immune response after vaccination is not normally evaluated but we considered a highlight for a correct and effective adaptive immune response. As we agree that it is important to explore the dynamics of cytokine induction throughout immunization, we will analyze different cytokines at different timepoints in the following experiments.

Q3. Why are challenge studies not being conducted to determine the effectiveness of vaccines in preventing the disease?

Response

A challenge study is the gold standard for vaccine efficacy assessment. Therefore, we are looking for an opportunity to analyze protective efficacy through challenge studies in a collaboration. In this first approach we wanted to observe the response and the effects of immunization with the peptides. In addition, we are characterizing different vaccine formulations and setting up the viral strains for the challenge.

Round 2

Reviewer 1 Report

Comments and Suggestions for Authors

The issues I am concerned about has been answered, its ok.